# Ex Vivo and In Silico Approaches of Tracheal Relaxation through Calcium Channel Blockade of 6-Aminoflavone and Its Toxicological Studies in Murine Models

**DOI:** 10.3390/biomedicines11071870

**Published:** 2023-06-30

**Authors:** Angélica Flores-Flores, Samuel Estrada-Soto, César Millán-Pacheco, Blanca Bazán-Perkins, Rogelio Hernández-Pando, Maximiliano Ibarra-Barajas, Rafael Villalobos-Molina

**Affiliations:** 1Departamento de Inmunofarmacología, Instituto Nacional de Enfermedades Respiratorias, Mexico City 14080, Mexico; perkins@unam.mx; 2Tecnológico de Monterrey, Escuela de Medicina y Ciencias de la Salud, Mexico City 14380, Mexico; 3Facultad de Farmacia, Universidad Autónoma del Estado de Morelos, Cuernavaca 62209, Mexico; cmp@uaem.mx; 4Instituto Nacional de Ciencias Médicas y Nutrición Salvador Zubirán, Mexico City 14080, Mexico; rhdezpando@hotmail.com; 5Unidad de Biomedicina, Facultad de Estudios Superiores-Iztacala, Universidad Nacional Autónoma de México, Tlalnepantla 54090, Mexico; maxibarrab@hotmail.com (M.I.-B.); villalobos@unam.mx (R.V.-M.)

**Keywords:** 6-aminoflavone, asthma, calcium channel blockade, toxicological studies, relaxant effect

## Abstract

Asthma is a condition in which a person’s airways become inflamed, narrowed, and produce greater amounts of mucus than normal. It can cause shortness of breath, chest pain, coughing, or wheezing. In some cases, symptoms may be exacerbated. Thus, the current study was designed to determine the mechanism of action of 6-aminoflavone (6-NH_2_F) in ex vivo experiments, as well as to determine its toxicity in acute and sub-chronic murine models. Tissues were pre-incubated with 6-NH_2_F, and concentration–response curves to carbachol-induced contraction were constructed. Therefore, tracheal rings pre-treated with glibenclamide, 2-aminopyridine, or isoproterenol were contracted with carbachol (1 µM), then 6-NH_2_F relaxation curves were obtained. In other sets of experiments, to explore the calcium channel role in the 6-NH_2_F relaxant action, tissues were contracted with KCl (80 mM), and 6-NH_2_F was cumulatively added to induce relaxation. On the other hand, tissues were pre-incubated with the test sample, and after that, CaCl_2_ concentration–response curves were developed. In this context, 6-NH_2_F induced significant relaxation in ex vivo assays, and the effect showed a non-competitive antagonism pattern. In addition, 6-NH_2_F significantly relaxed the contraction induced by KCl and CaCl_2_, suggesting a potential calcium channel blockade, which was corroborated by in silico molecular docking that was used to approximate the mode of interaction with the L-type Ca^2+^ channel, where 6-NH_2_F showed lower affinity energy when compared with nifedipine. Finally, toxicological studies revealed that 6-NH_2_F possesses pharmacological safety, since it did not produce any toxic effect in both acute and sub-acute murine models. In conclusion, 6-aminoflavone exerted significant relaxation through calcium channel blockade, and the compound seems to be safe.

## 1. Introduction

Asthma is one of the major diseases around the world, and studies about this ailment are minor because of its complex understanding, since it involves different immune pathways, a complex pharmacology, and shares a relationship with different respiratory illnesses [1,2,3]. The Global Initiative for Asthma (GINA) mentioned that this disease causes symptoms such as wheezing, shortness of breath, chest tightness, and coughing. These symptoms are associated with variable expiratory airflows, such as difficult gas exchange in the capillaries of the lungs due to bronchoconstriction, airway wall thickening, and the production of extra mucus [3]. Thus, treatment of these diseases must be based on bronchodilators and anti-inflammatory agents, although not all drugs are efficient, or they could cause adverse effects. In this context, the discovery and development of new drugs with a multitarget mode of action and with fewer adverse effects to treat asthma are required.

Recent research based on natural products has focused on flavonoids and flavonoid-like molecules due to their therapeutic benefits. Flavonoids are polyphenolic compounds with a characteristic chemical structure, and they are classified into flavones, flavanones, flavonols, anthocyanidins, flavanols, aurones, furan chromones, isoflavanones, biflavones, chalcones, and dihydrochalcones [4,5]. Due to the wide range of biological activities of flavones, a great interest has been generated for the discovery of active agents for the treatment of respiratory diseases, since flavones are one of the subclasses of flavonoids with diverse pharmacological properties, including anti-tuberculosis, antimicrobial, antioxidant, anti-inflammatory, and anti-allergic, among others [5]. Several flavones have relaxant effects on airway smooth muscle (ASM) in different ex vivo murine models [6,7,8]. In addition, flavones were reported with significant anti-inflammatory effects by the inhibition of cytokines, such as tumor necrosis factor-α, interleukin-1ß, nitrite ion free-radical generation inhibition, and lipid peroxidation inhibition [9]. Other interventions with flavonoids improved the health of some diseases with inflammatory processes [10]. Thus, those studies described all relaxant and anti-inflammatory effects of flavonoids, especially flavones, and suggest that these natural compounds could be developed as potential efficient anti-asthmatic drugs, due to their dual mode of action.

The current investigation aimed to determine the tracheal relaxation mode of action of 6-NH_2_F in ex vivo and in silico studies, and to determine its safety in pharmacological uses, through a toxicological approach in mice models based on OECD guidelines.

## 2. Materials and Methods

### 2.1. Materials

All chemicals were ACS-grade; 6-aminoflavone, carbachol, glibenclamide, 2-aminopyridine, theophylline, and isoproterenol were purchased from Sigma-Aldrich Chemical Co. with manufacture (St. Louis, MO, USA). Potassium chloride (KCl) and calcium chloride (CaCl_2_) were obtained from Merck (Germany/Darmstadt). For the ex vivo experiments, all compounds were dissolved in DMSO (1%) and then diluted with distilled water.

### 2.2. Animals

Male ICR mice (weighing 27–30 g) and male Wistar rats (weighing 250–300 g) were obtained from the “Instituto Nacional de Enfermedades Respiratorias (INER)” Bioterium. The animals were maintained under filtered, air-conditioned, 12/12 h light/dark cycles at 21 ± 1 °C and at 50–70% humidity; food and water were provided ad libitum. All the animals’ procedures were conducted in accordance with our Federal Regulations for Animal Experimentation and Care (SAGARPA, NOM-062-ZOO-1999) and with the international rules of care and use for laboratory animals. The protocol was approved by the Scientific and Bioethics Committee of the “INER” (on 1 April 2020) with assigned code B11-20.

### 2.3. Data Acquisition and Analysis System for Isolated Tissues

#### 2.3.1. Tracheal Rings Preparation

Rat tracheal rings were dissected, cleaned, and mounted in a 10 mL organ bath containing Krebs–Henseleit solution and maintained at 37 ± 0.5 °C with an isometric tension of 2 g, as described in [11]. In all experiments, contraction responses were measured and recorded using Grass-FT03 force transducers Astromed^®^ (West Warwick, RI, USA), connected to an MP100 analyzer BIOPAC^®^ Instruments (Santa Barbara, CA, USA). The tissues were equilibrated for at least 1 h prior to the addition of the test samples, and during this time, the buffer solution was renewed every 20 min before starting the assays (Appendix A, Figure A1).

#### 2.3.2. Functional Relaxant Effect

After a 30 min washout period, the tracheal rings were contracted with carbachol (CCh) (1 µM) twice at 30 min intervals and washed with Krebs–Henseleit solution. To determine the relaxant effect, after stabilization, the tissues were contracted again with submaximal contraction of CCh (1 µM), and once the plateau was attained, concentration–response curves of 6-NH_2_F, positive control (theophylline), and vehicle (DMSO 1%) were obtained by adding cumulative concentrations of the test samples to the bath [11,12].

#### 2.3.3. Functional Mechanism of Action Determination

To explain the relaxant effect of 6-NH_2_F, different possible mechanisms of action strategies on tracheal rings were evaluated, which are described in Table 1 [11,12].

### 2.4. In Silico Studies

Molecular docking of 6-NH_2_F was performed using L-type voltage-gated calcium channel from rabbit (*Oryctolagus cuniculus*) [13] (PDB ID: 6JPA, 6JPB, 6JP5, and 6JP8 [14]). *O. cuniculus* has a 92.63% sequence homology vs. its corresponding human homologue and 92.1% vs. *Rattus norvegicus* (Wistar rats), as reported in [15]. Based on the high sequence homology with *O. cuniculus*, we assume that all conclusions here stated might be applied to human or to Wistar rats where experiments were conducted. 6JP5 has a nifedipine binding site, so it has been used as reference/validation for this study. The 6-NH_2_F was drawn, converted to three-dimensional structure, and its best conformer generated with Marvin Sketch 21.10. Autodock Vina (USA) [16] was used for molecular docking due to its great capability to reproduce the binding modes of small molecules to a protein with high accuracy. All initial structures and configuration files were generated with Chimera UCSF [17] using the Autodock Vina plugin as implemented. All structures were structurally aligned to use the same box and size for all four structures. The box center was placed at the coordinates (163.78, 187.11, 172.58) with a size of (20 × 20 × 20). The box dimensions used in this study were enough to consider other possible binding sites distinct from nifedipine. A hundred independent molecular docking assays were performed on each calcium channel structure with both ligands. The results were clustered using a 2Å RMSD, and molecular interactions of the most popular cluster were analyzed on Maestro from Schrodinger [18]. All images were made using VMD or Maestro from Schrodinger (USA) [19].

### 2.5. Toxicological Studies

#### 2.5.1. Acute Oral Toxicity Test

The toxicity study was carried out according to the Organization of Economic Co-operation and Development (OECD) of acute oral toxicity with modification [20]. Five groups of five ICR mice each were administered with different doses of 6-NH_2_F (5, 50, 300, 1100, and 2000 mg/kg) orally through a cannula. After administration, the signs of toxicity and mortality were observed, such as changes in skin and fur, eyes, and mucous membranes, as well as the respiratory, circulatory, autonomic, and central nervous systems and somatomotor function, behavioral patterns, tremors, convulsions, drooling, diarrhea, lethargy, sleep, and coma; after that period, every 2 h until 24 h and over 14 days, animals were observed.

#### 2.5.2. Sub-Acute Oral Toxicity Test

The animals were weighed and divided into three groups (sham, vehicle, and 6-NH_2_F). The group treated with 6-NH_2_F had the compound orally administered for 28 days at 100 mg/kg per day; the sham group received physiological saline solution daily for 28 days; and the vehicle group were given a solution of tween 80 3% and DMSO 2%, administered orally, according to the guide of a repeated-dose 28-day oral toxicity study in rodents [14]. Body weight changes were supervised weekly for 28 days. The animals were monitored for manifestation of toxicity and mortality [21].

At the end of the test period, the animals of each group were anesthetized with sodium pentobarbital solution injection by intraperitoneal route. Blood samples were collected by cardiac puncture to test the transaminases activity (ALT and AST). The vital organs were recovered for histology studies (kidney, liver, heart, and lungs) using a formalin solution. Finally, a complete analysis of each animal with treatment was compared with the sham and vehicle animals and stained with hematoxylin and eosin. Histological slides were examined with a conventional light microscope (Zeiss 20×).

### 2.6. Statistical Analysis

Pharmacological data were analyzed by two-way analysis of variance, and *p* < 0.05 was considered statistically significant. All values are expressed as the mean ± standard error of the mean. Concentration–response curves were plotted, and the experimental data obtained were adjusted by means of a nonlinear curve-fitting program (ORIGIN 8.5).

## 3. Results and Discussion

In a previous study, we demonstrated that a series of structural-related flavonoids showed significant tracheal relaxant effect, and after a SAR study, 6-hydroxyflavone (6-OHF) and 6-NH_2_F were the most active compounds of flavonoids studied [8]. Thus, 6-OHF was used to determine its mechanism of action, which acts as a calcium channel blocker and an anti-asthmatic compound [11]. Based on those previous studies, in the current manuscript, we decided to explore the mechanism of action of 6-aminoflavone, which was the second-most potent compound of the series explored. Thus, 6-NH_2_F showed significant relaxation on isolated rat tracheal rings, demonstrating a concentration-dependent pattern with a relaxant effect of 91.1 ± 2.1%, compared with theophylline, a powerful relaxant agent of the smooth muscle fiber, bronchi, and peripheral vessels, which is used as a bronchodilator (Figure 1). The relaxing action of 6-NH_2_F confirmed the association with the relaxant activity on the airway smooth muscle reported for numerous flavonoids, especially flavanones and flavones [11,22]. Therefore, we decided to explore the experimental evidence on possible relaxant mechanism(s), and the following assays were focused on studying the airway tissues and the molecular signaling pathways involved.

First, in the adrenergic pathway, β_2_-adrenoceptors are distributed in airway smooth muscle, in epithelial, and in endothelial cells. The activation of these receptors is associated to the α- subunit of the Gs protein, together with a molecule of guanosine triphosphate (GTP) [13]; β_2_-adrenoreceptor activation increases the intracellular cAMP that activates protein kinase A, which in turn phosphorylates key regulatory contractile proteins and inhibits calcium ion release from intracellular stores, reducing membrane calcium entry and conducing tracheal relaxation (GINA, 2021) [3]. Consequently, to corroborate whether 6-NH_2_F induced its relaxant action by direct agonism of β_2_-adrenoreceptor and/or by activation of this via, rat tracheal rings were pre-incubated with isoproterenol (a β_1/2_-adrenoceptors agonist), and the relaxant effect of cumulative concentrations of 6-NH_2_F was determined on the incubated tissues. No significant modification of the relaxant curve was observed, indicating that this via is not implicated in the effect of the flavonoid assayed (Figure 2).

Then, it was necessary to explore other possible ways of the relaxant effect produced; thus, the antagonistic effect on cholinergic receptors was studied. Muscarinic receptors are important in pathophysiology of asthma, since when they are activated by acetylcholine/carbachol, they induce contraction in smooth muscle by increasing cytosolic Ca^2+^ through its release from internal stores, such as sarcoplasmic reticulum, and external stores, such as voltage-dependent influx channels [13,23]. In Figure 3, the effect of 6-NH_2_F is shown, which opposed contraction at different concentrations with a partial decrease of efficacy and the displacement to the right of the curves, suggesting that 6-NH_2_F showed a non-competitive antagonism provoked by an allosteric interaction with the receptor or/and through a functional antagonism acting on other molecular targets that decrease the contractile effect of carbachol, such as increased production of second messengers such as NO, a possible interaction with ion channels, or ryanodine receptor channels that have an important role in airway smooth muscle Ca^2+^ homeostasis [24].

Calcium, as an intracellular second messenger, plays different roles in the airways’ smooth muscle, where voltage-gated calcium channels and receptor-operated calcium channels do exist [5,25]. When these channels are open, intracellular Ca^2+^ concentration increases, and contraction of smooth muscle is generated. In this sense, to identify the role of 6-NH_2_F on voltage-dependent Ca^+2^ channels, rat tracheal rings were contracted by depolarization using KCl (80 mM), and the concentration–response curve of 6-NH_2_F was obtained to account for the external calcium entry to cells for them to contract. In Figure 4, it is observed that 6-NH_2_F relaxed the contraction induced by KCl (80 mM) such as nifedipine, a calcium channel blocker employed as a positive control. Moreover, to confirm this effect, the tissues were maintained in a Ca^2+^-free solution, and after 6-NH_2_F treatment, contraction was induced with cumulative concentrations of Ca^2+^. Figure 5 shows that the CaCl_2_-induced contraction was significantly reduced with different concentrations of 6-NH_2_F (75, 134, and 209 μM), suggesting that calcium channel blockade is the main mechanism of action involved in its relaxant action. It is important to mention that these channels play a significant role in the contraction of the respiratory tissues involved in asthma, due to the Ca^2+^-free concentrations in the cytoplasm [26]. Furthermore, Ca^2+^ regulates many aspects of cell function from mediators, neurotransmitters, hormones, mast cell mediators, and mucus secretions, among others [23].

To finish establishing the functional mechanism of action of the flavonoid, it was decided to incubate tissues with glibenclamide and 2-aminopyridine (Figure 6), two potassium channel blockers; however, the relaxant effect of 6-NH_2_F with these two substances was not modified, which indicated to us that the potassium channel opening was not involved.

All of these results indicate that the mechanism of action of 6-NH_2_F is based mainly on calcium channel blockade; thus, to complement the finding in functional studies, an in silico approach was conducted to predict the possible interactions of 6-NH_2_F within Ca^2+^ channels. For this, the flavonoid and nifedipine were docked on four different structures of L-type calcium channel from *O. cuniculus*. Multiple docking assays (100 repetitions each) were carried out to improve molecular sampling on those structures. Table 2 summarizes the results obtained in this work. As noted, 6-aminoflavone showed lower-affinity energy when compared with nifedipine on all structures. On the other hand, nifedipine molecular docking reproduced the crystallographic conformation only in 54% of the assays. This can be explained due to the presence of a lipid molecule on the crystallographic structure that was not used in this experiment, as reported in [27].

Two possible binding sites were found for 6-NH_2_F in this study, which are located close to those found in the crystallographic ones. One of them was found exclusively on 6JPA structure as noted on interaction maps (Figure 7), and the other was found in the same region on 6JPB, 6JP5, and 6JP8. The 6JPA binding site was constituted by L566, L563, F608, L611, T612, N649, L652, F656, F1013, I1050, A1053, and F1054. Two residues of this binding site were unique for 6-NH_2_F (L563 and L566); the other ones were residues also found for verapamil. As noticed, the 6JPA binding site was composed primarily of hydrophobic residues. There exist hydrophilic residues (T612 and N649) on this binding site, located near the amino group of the ligand and interacting with it by hydrogen bonds (T612).

Another possible binding site was located on the other three structures, and it shared seven residues: V932, T1012, M1057, F1060, Y1365, A1369, and I1373. This binding site is where the crystallographic ligands (diltiazem, nifedipine, and agonist Bay K 8644) are bound on their corresponding structures. As noticed in Figure 7a, F1060 was interacting with 6-NH_2_F through π–π interactions on the three structures. Y1365 was found interacting with 6-NH_2_F by hydrogen bond on 6JP8, exclusively. There exist two hydrophilic residues on this binding site, and one of them was located near the amino group of 6-NH_2_F. Figure 7b shows binding sites found in this work. Nifedipine interaction maps were reported previously [27]. With this study, we verified that 6-NH_2_F strongly interacts with its possible binding site as a calcium channel blocker. Thus, all these results can offer evidence to develop a specific airway relaxant agent for the potential treatment of asthma; however, it is necessary to authenticate these pharmacological, functional, and in silico findings by using an in vivo model of asthma. In the meantime, it is necessary to evaluate the pharmacological safety uses, and acute and sub-chronic toxicity assays can bring us closer to the final goal, which is to develop an effective and safe anti-asthmatic agent.

Therefore, in the acute study, 6-NH_2_F did not exhibit any apparent physical changes or death at different doses (5, 50, 300, and 2000 mg/kg), after 5 and 24 h observation, and for the next 14 days. The medium lethal dose (LD_50_) of 6-NH_2_F is over 2000 mg/kg, and according to the criteria of the Globally Harmonized System (GHS), it is classified in category 4. Moreover, 6-NH_2_F was tested with a unique dose of 100 mg/kg for 28 days compared with vehicle and sham groups, and during that period, each group was monitored for changes in weight. In Figure 8, the percentage of body weight variation of all three groups is shown; as observed, there were no changes with a normal increase as time went by, with respect to food and water consumption. The results confirmed that 6-NH_2_F is not a hazardous product. In addition, Figure 9 shows the relative weight of the main organs dissected from assayed mice, such as heart, kidney, lung, and liver, while Figure 10 shows transaminases’ activities (alanine aminotransferase and aspartate aminotransferase) after blood was obtained by heart puncture, indicating that metabolism and liver function was not altered after 28 days of treatment.

Finally, to confirm that 6-NH_2_F did not produce any toxicological sign after administration of 100 mg/kg for 28 days, histopathological examinations of the heart, kidney, lung, and liver were made, obtaining that the test sample did not cause any pathological damage or cell alterations of tissue assayed when compared with vehicle and sham mice groups, proving that 6-NH_2_F did not show necrosis, change in the structure of cells, replication, or loss of cell nucleus that indicate damage (Figure 11).

These results together provide an important insight into the possible mechanism of relaxant action and pharmacological safety in its uses as a possible future anti-asthmatic drug. Moreover, it is important to develop studies from the immunology point of view because asthma is considered a complex disease, which involves inflammation; then, it is necessary to study the possible anti-inflammatory properties of 6-NH_2_F on the airway, as an additional proposed treatment to control cellular infiltration that plays an important role in the pathogenesis of asthma.

## 4. Conclusions

Our results suggest that 6-NH_2_F has a significant relaxant activity by calcium channel blockade, interfering in the contraction of smooth muscle in the airways as the main mode of action. Additionally, a 6-NH_2_F molecular docking study revealed that it binds to the L-type calcium channel with better affinity energy than nifedipine, which is a drug with a potent L-type calcium channel-blocking effect. Thus, with all these results 6-NH_2_F can be proposed as a potential agent for the treatment of asthma and related respiratory diseases.

## Figures and Tables

**Figure 1 biomedicines-11-01870-f001:**
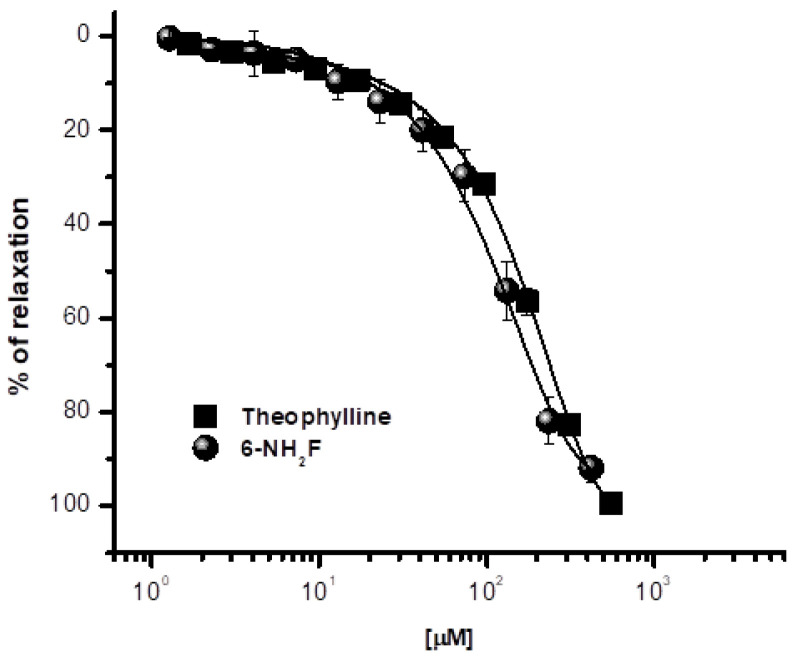
Relaxant concentration–response curves of 6-aminoflavone (6-NH_2_F) on carbachol-contracted tracheal rings. Each point represents the mean ± S.E.M. of 5 animals.

**Figure 2 biomedicines-11-01870-f002:**
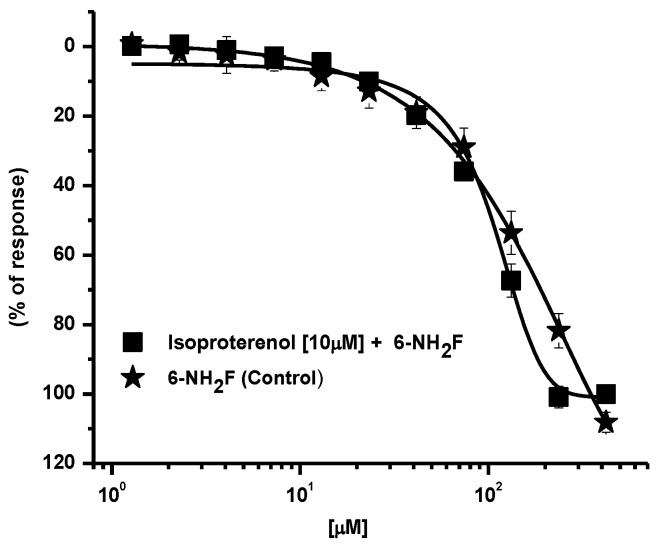
Relaxant concentration–response curves of 6-NF_2_H on carbachol-contracted tracheal rings, pre–incubated with isoproterenol. Each point represents the mean ± S.E.M. of 5 animals.

**Figure 3 biomedicines-11-01870-f003:**
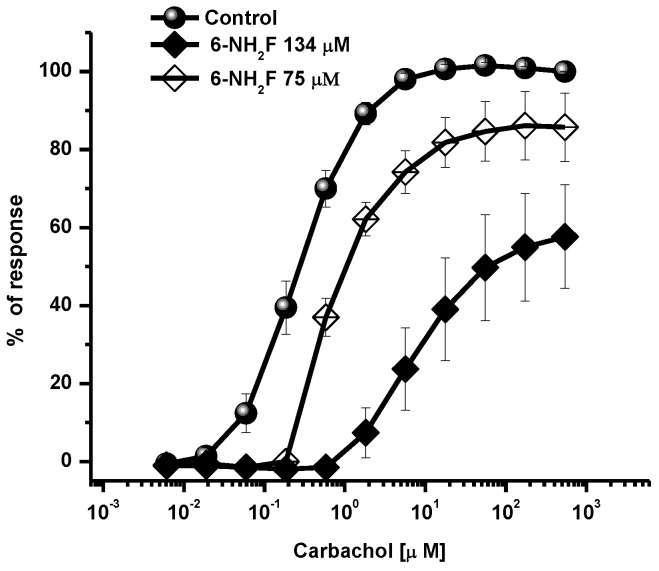
Concentration-response curves of carbachol-contracted tracheal rings in the presence of a different concentration of 6-NH_2_F. Each point represents the mean ± S.E.M. of 5 animals.

**Figure 4 biomedicines-11-01870-f004:**
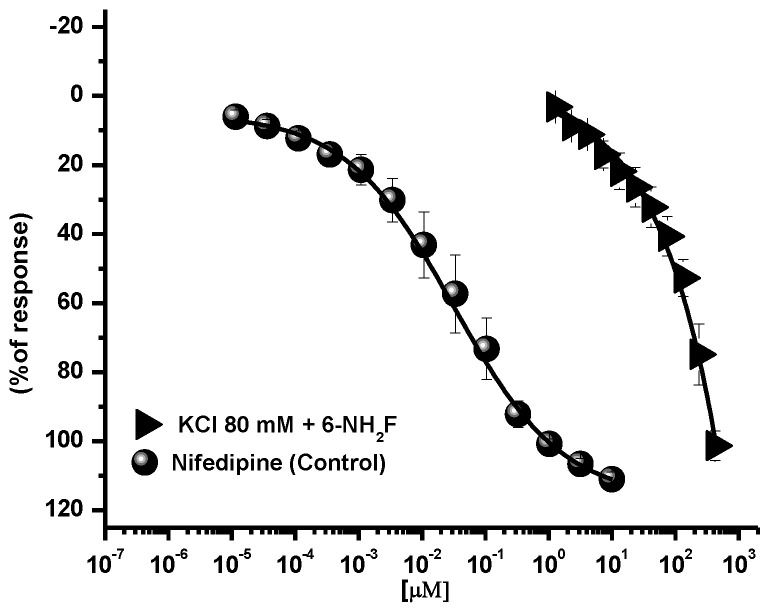
Concentration-response curves of the relaxant effect of 6-NH_2_F on the contraction induced by KCl (80 mM) in tracheal rings. Each point represents the mean ± S.E.M. of 5 animals.

**Figure 5 biomedicines-11-01870-f005:**
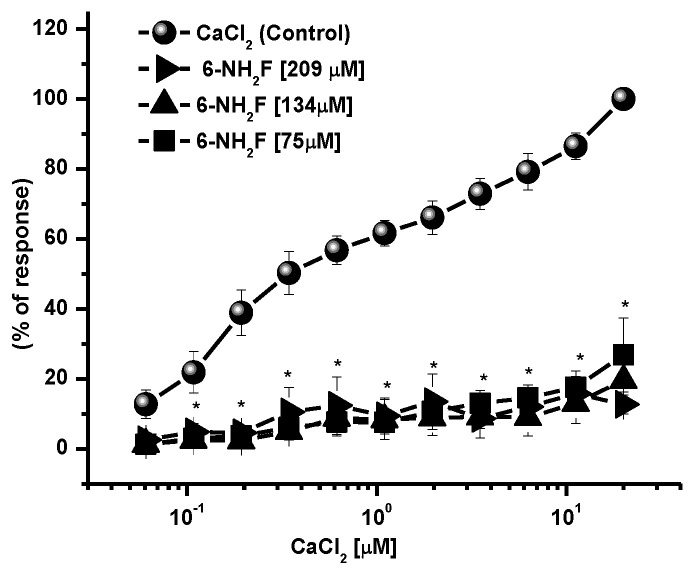
Concentration–response curves of 6-NH_2_F on the contraction induced by cumulative concentrations of CaCl_2_, in Ca^2+^-free Krebs–Henseleit solution. Each point represents the mean ± S.E.M. of 5 animals. * *p* < 0.05 control vs. 6-NH_2_F.

**Figure 6 biomedicines-11-01870-f006:**
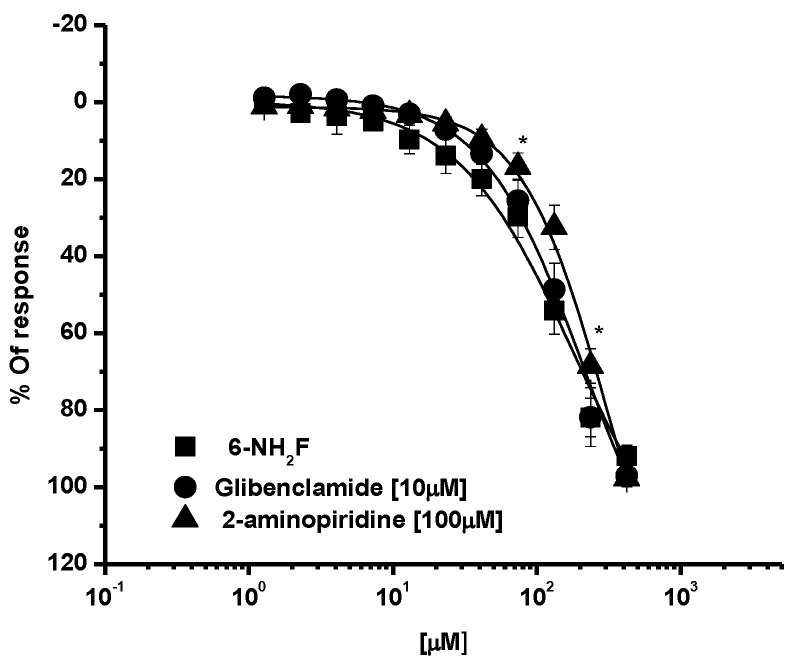
Relaxant concentration-response curves of 6-NH_2_F in the presence of glibenclamide (10 μM) or 2-aminopyridine (100 μM) in tracheal rat rings. Each point represents the mean ± S.E.M. of 5 animals. * *p* < 0.05 6-NH_2_F vs. glibenclamide and 2-aminopiridine.

**Figure 7 biomedicines-11-01870-f007:**
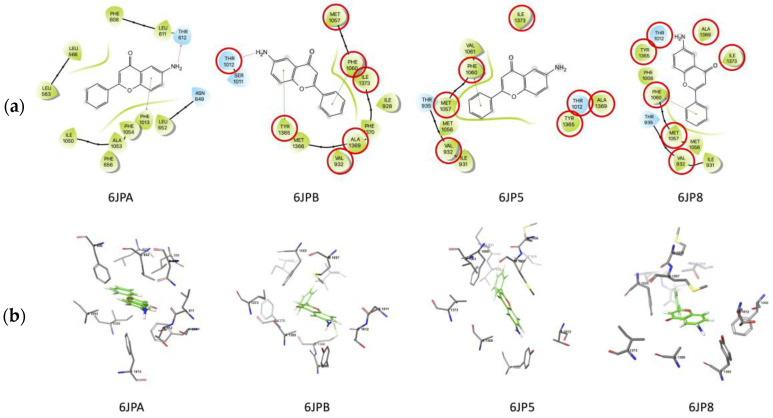
(**a**) Molecular docking of 6-NH_2_F and (**b**) nifedipine on L-type calcium channel. The red circles represent of pharmacophoric interactions.

**Figure 8 biomedicines-11-01870-f008:**
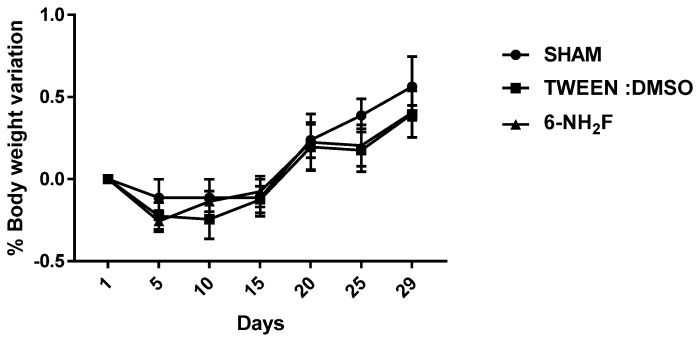
Body weight of mice treated with 6-NH_2_F (100 mg/kg) for 28 days. Each bar represents the mean ± the standard error of the mean, n = 5 animals.

**Figure 9 biomedicines-11-01870-f009:**
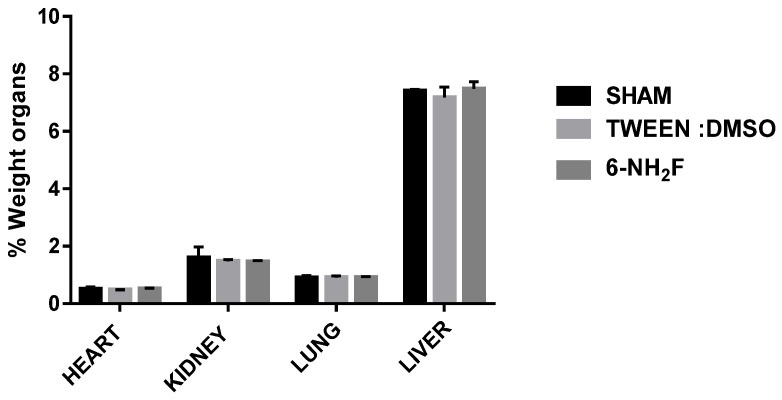
The relative weight of the heart, kidney, lung, and liver was obtained for different groups after 28 days of treatment with 6-NH_2_F. Each bar represents the mean ± standard error of the mean, n = 5 animals.

**Figure 10 biomedicines-11-01870-f010:**
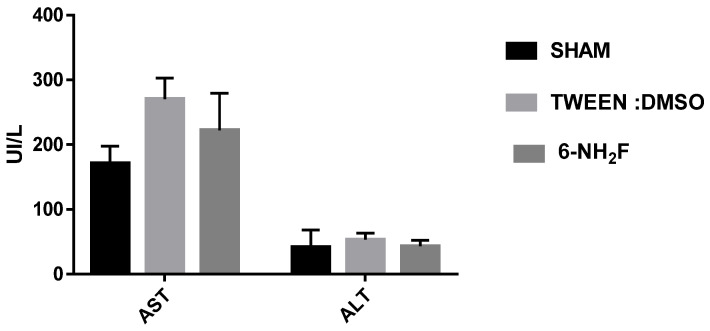
Blood transaminases were determined in animals treated daily with 6-NH_2_F (100 mg/kg) for 28 days, vehicle and sham. Each bar represents the mean ± the standard error of the mean, n = 5 animals.

**Figure 11 biomedicines-11-01870-f011:**
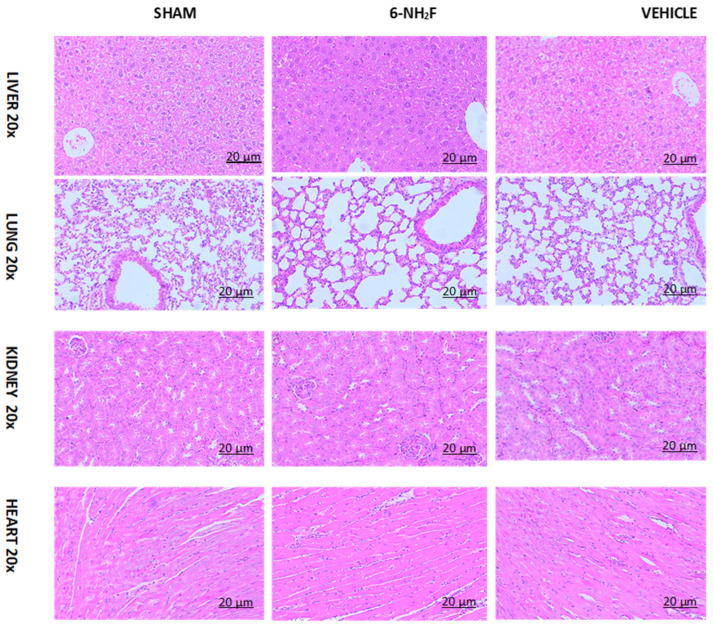
Micrography of histology of lung, heart, kidney, and liver of mice treated for 28 days with repeated administration of 6-aminoflavone (6-NH_2_F) (100 mg/kg), vehicle (DMSO 2%: Tween 80 10%), and sham group (water).

**Table 1 biomedicines-11-01870-t001:** Protocol of the study and methods of evaluating various mechanisms of the relaxant effect of 6-aminoflavone [11,12].

Mechanism Explored	Contractile Agent	Conditions and Strategy	Incubating Substance
Muscarinic receptor antagonism	Carbachol (a cholinergic agonist)	Incubate tissues with 6-NH_2_F (75 and 134 µM) in normal Krebs–Henseleit solution; after that, concentration–response curves to carbachol were constructed.	EC_50_ of 6-NH_2_F
Calcium channel blockade	KCl (80 mM)	After tissues contraction with KCl in a normal Krebs–Henseleit solution, concentration–response relaxation curves with 6-NH_2_F or nifedipine (calcium channel blocker) were obtained.	EC_50_ of 6-NH_2_F Nifedipine (1 µM)
Calcium channel blockade	CaCl_2_	Tissues were incubated in calcium-free Krebs–Henseleit solution and pre-treated with 6-NH_2_F (75, 134, and 209 µM); later, concentration–response curves of CaCl_2_ were constructed.	Different concentrations of 6-NH_2_F
Potassium channel opening	Carbachol (1 µM)	Before tissues contraction with carbachol in normal Krebs–Henseleit solution, tracheal rings were pre-incubated (15 min) with potassium channel blockers, then concentration–response relaxation curves with 6-NH_2_F were obtained.	Glibenclamide (10 µM) 2-aminopiridyne (100 µM)
β-adrenoceptor stimulation	Carbachol (1 µM)	Before tissues contraction with carbachol in normal Krebs–Henseleit solution, tracheal rings were pre-incubated (15 min) with isoproterenol, then concentration–response relaxation curves with 6-NH_2_F were obtained.	Isoproterenol (10 µM)

**Table 2 biomedicines-11-01870-t002:** Nifedipine and 6-aminoflavone affinity energies (kcal/mol) were obtained in this work.

	6JPA	6JPB	6JP5	6JP8
Nifedipine	−5.90 ± 0.02 (100)	−6.79 ± 0.04 (98)	−6.86 ± 0.14 (54)	−7.50 ± 0.03 (98)
6-aminoflavone	−7.78 ±0.15 (87)	−7.30 ±0.00 (93)	−7.50 ± 0.00 (100)	−7.28 ± 0.04 (98)

Numbers in parentheses are member percentages on the most popular cluster used for analysis.

## Data Availability

The data presented in this study are available upon request from the corresponding author.

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
