# Peer review of "Ex Vivo and In Silico Approaches of Tracheal Relaxation through Calcium Channel Blockade of 6-Aminoflavone and Its Toxicological Studies in Murine Models"

_biomedicines, 2023, doi:10.3390/biomedicines11071870_

Round 1
Reviewer 1 Report
In this manuscript, Flores-Flores, Estrada-Soto and their co-workers reported their findings on the functional mechanism of tracheal relaxation of 6-aminoflavone (6-NH2F). The possible mechanism of action studied therein included b2 adrenergic receptor agonists, muscarinic receptor antagonists, potassium channel blockers, and calcium channel blockers. In addition, computer modelling was performed to investigate the possible interactions of 6-NH2F within Ca+2 channels. The target compound 6-NH2F was docked on four different structures of the L-type calcium channels and compared with the docking results of the reference compound nifedipine, which is a calcium channel blocker medication. Finally, the acute and sub-chronic toxicity of the target compound were carried out to evaluate its safety. According to their ex vivo and in silico results, 6-NH2F exerted the relaxant effect through calcium channel blockade. Moreover, the toxicological data shows that 6-NH2F exhibited pharmacological safety.
In 2017 (ref. 8 in this manuscript, Med. Chem. Res. 2018, 27, 22−7), the authors reported the relaxant effect on the tracheal rat rings of 10 flavonoids. Among these, 6-hydroxyflavone (6-HOF) was found to be the most active compound and 6-NH2F was the second. Then in 2018 (ref. 11 in this manuscript, Drug Dev. Res. 2019, 80, 218−29), the authors used 6-HOF as the target to study its mechanism of action and toxicology in murine models.
Though this manuscript has provided some sound results, it lacks of novelty based on the following reasons. The first is that the relaxant activity of the target compound (i.e., 6-NH2F) has already been published in Med. Chem. Res. 2018, 27, 22−7. The second is the approaches and methods, including ex vivo, in silico, and in vivo, used in this manuscript are similar to those applied to 6-HOF (published in Drug Dev. Res. 2019, 80, 218−29). The authors need to provide their justifications or reasons why they studied the functional mechanism of 6-NH2F as its relaxant activity was less than that of 6-HOF. Moreover, 6-NH2F and 6-HOF have the same position of the substituent on flavones; NH2 is an isostere of OH.
In short, this manuscript is not recommended for its publication in Biomedicines.
Overall, quality of English language for this manuscript is satisfactory. Moderate editing is suggested to be done by a native English speaker.
Author Response
Point 1: In this manuscript, Flores-Flores, Estrada-Soto and their co-workers reported their findings on the functional mechanism of tracheal relaxation of 6-aminoflavone (6-NH2F). The possible mechanism of action studied therein included b2 adrenergic receptor agonists, muscarinic receptor antagonists, potassium channel blockers, and calcium channel blockers. In addition, computer modelling was performed to investigate the possible interactions of 6-NH2F within Ca+2 channels. The target compound 6-NH2F was docked on four different structures of the L-type calcium channels and compared with the docking results of the reference compound nifedipine, which is a calcium channel blocker medication. Finally, the acute and sub-chronic toxicity of the target compound were carried out to evaluate its safety. According to their ex vivo and in silico results, 6-NH2F exerted the relaxant effect through calcium channel blockade. Moreover, the toxicological data shows that 6-NH2F exhibited pharmacological safety.
In 2017 (ref. 8 in this manuscript, Med. Chem. Res. 2018, 27, 22−7), the authors reported the relaxant effect on the tracheal rat rings of 10 flavonoids. Among these, 6-hydroxyflavone (6-HOF) was found to be the most active compound and 6-NH2F was the second. Then in 2018 (ref. 11 in this manuscript, Drug Dev. Res. 2019, 80, 218−29), the authors used 6-HOF as the target to study its mechanism of action and toxicology in murine models.
Though this manuscript has provided some sound results, it lacks of novelty based on the following reasons. The first is that the relaxant activity of the target compound (i.e., 6-NH2F) has already been published in Med. Chem. Res. 2018, 27, 22−7. The second is the approaches and methods, including ex vivo, in silico, and in vivo, used in this manuscript are similar to those applied to 6-HOF (published in Drug Dev. Res. 2019, 80, 218−29). The authors need to provide their justifications or reasons why they studied the functional mechanism of 6-NH2F as its relaxant activity was less than that of 6-HOF. Moreover, 6-NH2F and 6-HOF have the same position of the substituent on flavones; NH2 is an isostere of OH.
In short, this manuscript is not recommended for its publication in Biomedicines.
Comments on the Quality of English Language
Overall, quality of English language for this manuscript is satisfactory. Moderate editing is suggested to be done by a native English speaker..
Response 1: As mentioned by the Reviewer 1, all strategy was developed to determine the functional mechanism of action of 6-aminoflavone, which is a structural analogue of 6-hydroxy-flavone. Furthermore, both compounds (6-NH2F and 6-OHF) were the two most active compounds of the series of ten that were evaluated in a previous study (ref. 11 in this manuscript, Drug Dev. Res. 2019, 80, 218− 29);, even though they are isosteres this does not guarantee that both compounds could have the same mechanism of action (since they are structural analogues), since a small structural change could generate a dramatic change in the biological activity or in its mechanism of action, a term that is known as activity cliffs described by Medina-Franco et al. 2009. In this case, the amino group could be protonated and deactivated and thus not produce a pharmacological effect, or have a greater number of hydrogen bonding interactions compared to alcohol, which would cause other types of interactions and produce the effect through a different mechanism of action. That is why it is always necessary to evaluate the mechanism of action of the molecules that are most active in a pharmacological study carried out.
Medina-Franco et al., J Chem Inf Model 2009; 49:477-491.
Reviewer 2 Report
This is an interesting study. However, some problems could be identified:
-The title refers “in silico”, however in abstract no mention to this was included. In addition, as the general focus of this work is asthma, it is also strange to not find data on this subject in abstract.
-English should be improved in all document. Examples: “Current study was designed to determine the mechanism of action of 6-aminoflavone (6-NH2F) in ex vivo experiments; also, to determine its toxicity in acute and sub-chronic murine models.”; “bronchodilators and anti-inflammatory agents; although not all drugs are efficient or they could cause adverse effects.”
-6-Aminoflavone (6-19 NH2F) is the compound explored in this study. In a medicinal/pharmaceutical chemistry context, authors must explain clearly the selection if this molecule, among other similar structures, for the studies performed. IN addition, the results observed must also be discussed in this context.
-More details are needed in experimental section
-“Autodock Vina was used for molecular docking” – why the use of this software? This must be explained in the manuscript. In addition, docking validation??
-“2.4. In silico Pharmacological Studies” In my opinion, in silico studies are not pharmacology studies
-In toxicology tests “signs of toxicity and mortality were observed” – authors must explain in the manuscript which signs of toxicity were evaluated
-Table 2 data should be improved. For example, no mention to nifedipine is present
-English should be improved in all document. Examples: “Current study was designed to determine the mechanism of action of 6-aminoflavone (6-NH2F) in ex vivo experiments; also, to determine its toxicity in acute and sub-chronic murine models.”; “bronchodilators and anti-inflammatory agents; although not all drugs are efficient or they could cause adverse effects.”
Author Response
This is an interesting study. However, some problems could be identified:
-The title refers “in silico”, however in abstract no mention to this was included. In addition, as the general focus of this work is asthma, it is also strange to not find data on this subject in abstract.
Response 2: It is now included (lines 14-17): Also, 6-NH2F significantly relaxed the contraction induced by KCl and CaCl2, suggesting a potential calcium channel blockade, which was corroborated by in silico molecular docking that was used to approximate the mode of interaction with the L-type Ca2+ channel, where 6-NH2F showed lower affinity energy when compared with nifedipine. and also information about asthma (lines 1-4).
-English should be improved in all document. Examples: “Current study was designed to determine the mechanism of action of 6-aminoflavone (6-NH2F) in ex vivo experiments; also, to determine its toxicity in acute and sub-chronic murine models.”; “bronchodilators and anti-inflammatory agents; although not all drugs are efficient or they could cause adverse effects.”
Response: Done
-6-Aminoflavone (6-19 NH2F) is the compound explored in this study. In a medicinal/pharmaceutical chemistry context, authors must explain clearly the selection if this molecule, among other similar structures, for the studies performed. IN addition, the results observed must also be discussed in this context.
Response: In a previous study, which was cited in the manuscript (Flores-Flores et al., 2018), we demonstrated that a series of structural related flavonoids showed significant tracheal relaxant effect, and after a SAR study, 6-hydroxyflavone and 6-aminoflavone were the most active compounds of flavonoids studied. Thus, 6-hydroxyflavone were subjected for determine its mechanism of action, which also acts as calcium channel blocker and anti-asthmatic compound (Flores-Flores et al., 2019). Based on those previous studies, in current manuscript we decided to explore the mechanism of action of 6-aminoflavone, which was the second most potent compound of the series explored.
-More details are needed in experimental section
-“Autodock Vina was used for molecular docking” – why the use of this software? This must be explained in the manuscript. In addition, docking validation?
Response: Autodock Vina was used for molecular docking due to its great capability to reproduce the binding modes of small molecules to a protein with high accuracy as noted on Forli, Stefano, Ruth Huey, Michael E. Pique, Michel Sanner, David S. Goodsell, and Arthur J. Olson. “Computational Protein-Ligand Docking and Virtual Drug Screening with the AutoDock Suite.” Nature Protocols 11, no. 5 (May 2016): 905–19. https://doi.org/10.1038/nprot.2016.051.
Please refer to line 113 where we pointed out that nifedipidine on 6JP5 was used as reference for this study. Lines 243-246 and Table 1 showed our docking validation results on 6JP5 structure. We tested nifedipidine and 6-aminoflavone on four distinct structures including 6JP5 (where nifedipine was bound originally).
-“2.4. In silico Pharmacological Studies” In my opinion, in silico studies are not pharmacology studies
Response: you're right. It was changed by In silico studies.
-In toxicology tests “signs of toxicity and mortality were observed” – authors must explain in the manuscript which signs of toxicity were evaluated.
Response: For acute exposure, The first four hours (especially the first half hour) special attention was paid to observing signs and symptoms such as changes in skin and fur, eyes, and mucous membranes, as well as the respiratory, circulatory, autonomic and central nervous system and somatomotor function, behavioral patterns, tremors, convulsions, drooling, diarrhea, lethargy, sleep, and coma, after that period, every 2 h until 24 h, and for 14 days animals were observed as suggested by the OECD guide 420.
For subacute exposure, they were observed for 28 days daily and observed the same signs recommended in OECD guide 407.
In each of them, it was observed if there was mortality at 24h and during the 28 days.
-Table 2 data should be improved. For example, no mention to nifedipine is present
Response: Table 2 was improved.
Comments on the Quality of English Language
-English should be improved in all document. Examples: “Current study was designed to determine the mechanism of action of 6-aminoflavone (6-NH2F) in ex vivo experiments; also, to determine its toxicity in acute and sub-chronic murine models.”; “bronchodilators and anti-inflammatory agents; although not all drugs are efficient or they could cause adverse effects.”
Response: The English was improved.
Author Response
In the present work, the possibility of the involvement of the tracheal relaxant mode of action of 6-NH2F was investigated in ex vivo and in silico studies, and its safe pharmacological uses were investigated through a toxicological approach in animal models. The article is based on a perfect experimental design. All materials and methodology are excellently presented. Doses for acute and sub-acute oral toxicity are extremely adequately selected. Multi-organ histopathological studies demonstrate the low toxicity of 6-NH2F. All the obtained results suggest that flavone 6-NH2F could be considered as a new therapeutic agent for the treatment of asthma.
My questions;
What does the abbreviation OECD mean?
Response: Organization of Economic Co-operation and Development (the abbreviation and definition were included in the manuscript.
Shouldn't the table on page 3 have Number 1?
Response: Yes, it must be 1 (this was corrected in the manuscript).
Shouldn't the table on page 8 be Number 1?
Response: No, it must be 2 (this was corrected in the manuscript)
Round 2
Reviewer 2 Report
The document was improved and now it is more acceptable for publication.
Minor editing of English language required.